# Waiting in the Wings:
## Performance Interventions for Human Drone Interaction

| Zachary McKendrick | Ori Fartook | Patrick Finn | Ehud Sharlin | Jessica Cauchard |
|---|---|---|---|---|
| University of Calgary | Ben Gurion University of the Negev | University of Calgary | University of Calgary | Ben Gurion University of the Negev |

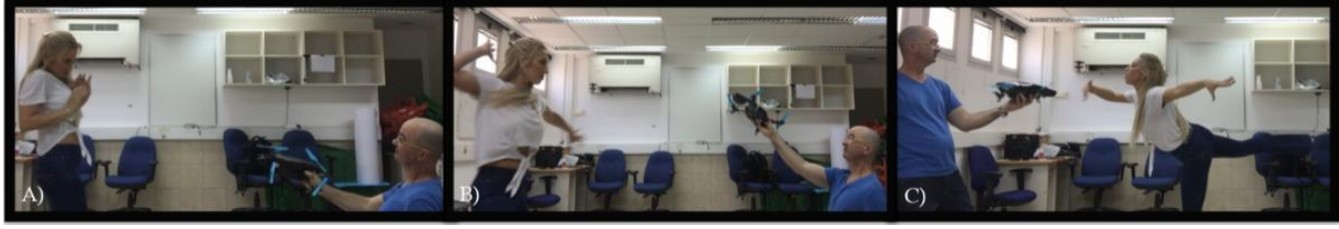

Figure 1: Performance Workshop exploring relationship between non-human (drone) and human actors. A) Drone in a low aloof position. B) Drone in a high aloof position. C) Drone in a high intimate position.

**ABSTRACT**

Drones inhabit live performance spaces from *Cirque du Soleil* to the Olympics. Can Human Drone Interaction (HDI) research learn from live performance? This paper positions HDI research within a formal performance framework. We build on existing HDI and performance collaborations through Laban Movement Analysis (LMA), and present findings from a Performance Workshop (PW) conducted with professional performers focusing on the unique expressive qualities of drones. Our findings expand on LMA and contribute a drone-specific movement lexicon and framework for use in HDI research that introduces *height* and *proximity* as key factors in nonverbal communication. Additionally, we explore live performance applications as a new design space and classification system for HDI research use cases. We suggest that performance approaches can result in more intentionally expressive drones that improve communication providing deeper connections with human actors on stage and with audience members.

**Keywords**: Performance Practice, Human Drone Interaction, Movement Based Interaction, Drama

**Index Terms**: K.6.1 [Management of Computing and Information Systems]: Project and People Management—Life Cycle; K.7.m [The Computing Profession]: Miscellaneous—Ethics

## 1 INTRODUCTION: THE HUMAN-DRONE RELATIONSHIP

Drones are often characterized as tools that only connect to humans through the completion of a task. In performance, developing work is showing the potential for human drone collaboration. While there is a dearth of research in the HDI literature related to arts, entertainment and drones, the performance industry was quick to adopt drones. Drones as lighting sources are an early use-case.

\* zachary.mckendrick@ucalgary.ca; orifar@post.bgu.ac.il; pfinn@ucalgary.ca; ehud@cpsc.ucalgary.ca; cauchard@post.bgu.ac.il

Light-up drones began replacing pyrotechnics in live performances, from the 2022 Olympics opening ceremony [9] to musical acts like *Metallica* [2]. Drones are also integrated directly into work as central figures in more demanding contemporary performances such as *Cirque du Soleil*'s *SPARKED* [45] or *ELEVENPLAY* and *Rhizomatiks*' multiple dance collaborations [3, 4] using drone choreography. While most light drones rely on Omni-directional color-changing LEDs to create moving pictures and aerial displays [32, 47, 56], collaborations between groups like Rhizomatiks and ELEVENPLAY have demonstrated the potential applications of directional lighting [4], capable of following, highlighting, and capturing movements and actions on stage. These performances show the potential of drones in live art. Once a drone and human occupy the same space a relationship is created between the drone and human inter-actor defined not only by our awareness of the flying robot's presence but also its proximity and the information it communicates through its actions. This paper explores the potential of the Human-Drone relationship in live performance.

When a drone enters a human space, its presence initiates a relationship. For example, prior research has shown that people are able to recognize a drone's flight path [51], movement and behavior [17], and even a drone's facial expressions [26] as conveying emotions, thus deeply impacting the interaction with it. Thus, our research is interested in exploring how a drone's unique attributes can influence relationships with bystanders and more specifically, direct human interactors. Viewing drones as public-facing interactive agents, we adopt a creative performance approach to investigate the potential of drones as dynamic and responsive inter-*actors*. To do so, this paper explores live performance scenarios as primary investigation site for Human Drone Interaction (HDI) through a brief and intensive performance exploration workshop with two professional dancers hosted by a professional theatre artist. The workshop was designed to answer our primary research question: *can we leverage live performance techniques to enhance existing movement practices already familiar to HDI?* Findings from our Performance Workshop (PW) are presented in this paper as an expanded framework of Laban Movement Analysis [1] for use in performance scenarios and HDI research. Additionally, we contribute technical theatre terms as a theoretical framework for the classification of drone use cases and broader deployment scenarios.

This paper outlines our performance centric approach to the HDI domain and seeks to answer our research question. First we

introduce some of the more prominent examples of where and how drones are being incorporated into live performances and how that has motivated our research. Next, we briefly address gaps in HDI exploration using findings from a recent meta review of current domain research and works that investigate interaction through physical movement. Then we introduce a new design space framework for drone deployment borrowed from live performance [23, 24] and organized using adapted HDI research [12]. We suggest that current drone applications also fit within the four performance categories we identify: *props, setting, FX*, and *performer* based on their function, autonomy, and level of interactivity. Then we reflect and expand on performance techniques utilized in previous HDI x Performance research collaborations, followed by a breakdown of our performance workshop and presentation of our findings and limitations. The result is a theoretical framework that uses existing relationships between performance and HDI to develop and introduce new movement techniques and vocabulary for experimentation with dynamic drone inter-actors on and off stage.

## 2 RELATED WORK

A recent scoping review of HDI publications identified "entertainment applications" as the second most cited drone research domain [27]. Included within this categorization are applications that range from game playing to racing. Of the ninety papers comprising the entertainment domain, only two works mention drama in any capacity [21, 22]. When expanding the traditional definition of entertainment to include movement, we find only four papers that use movement as a somatic practice [21, 22, 28, 37]. Indeed, many references to movement relate to drones' flight path and trajectory rather than movement practices. The evidence presented in the above scoping review suggests that despite increased interest and social integration, there is a lack of investigation into the application of drones in live performance scenarios in HDI research. This scoping review helps to illuminate current perspectives and priorities amongst HDI researchers and as drones become more affordable and accessible, we see them deployed in more mainstream public performance scenarios.

In the following section we explore HDI research that uses drones as direct interactive agents in experimental performance scenarios. We also introduce examples from HDI's research neighboring field human *robot* interaction (HRI) to provide a broader scope of how drama and performance practice has been applied across different form factors of non-autonomous agents. A performer is "a person who entertains an audience" [47], and an actor is "a participant in an action or process" [5]. We intentionally use the terms performer and actor interchangeably. Amalgamating performer and actor definitions from entertainment and research broaden participants' roles, giving them more agency to contribute to the creative process. Additionally, combining these terms allows us to apply performance principles to drones as non-humans that participate in entertaining an audience, whether on stage or in social interactions.

Having reviewed examples of performance practice and theory from both HDI and HRI, in subsection 2.2, we pivot our examination and introduce our performance framework. We breakdown the live performance space into four categories of classification based on their general level of autonomy dynamism and offer this classification system to the HDI community as a way to label the interactive potential (or intention) of their research. Each quadrant borrows from one of four main stage craft categories: setting, effects, props, and performers. In addition to defining each category we provide examples of how we would classify drone interactions using contemporary research.

## 2.1 Human and Non-human Actors

Very recently work has begun exploring the involvement of drones as a learning tool in performing arts scenarios. [55, 56]. Both examples use drones in a teaching capacity. Wilson-Small, et al. [55] found that haptic feedback (i.e., touch) from the drones was the preferred method of instruction among participants. They noted that haptic feedback was more effective at communicating correct physical movement and positioning to the performer than an incremental feedback approach [55]. Their conclusions are the result of a series of pilot studies with untrained participants, applying a drone's physical contact to correct their movement and positions. In this work, the drone adopts an authoritative role, and not that of a collaborator. However, drones have been used to inspire peoples' movement such as in the case of Drone-Chi [38]. In another study, Wilson-Small, et al. [56] created drone behaviors that motivated physical movement instead of correcting it through contact improvisation, resulting in a performance collaboration between the drone and professional dancer [56].

The past decade has seen an influx of research that places robots and drones in performance situations, replicating or replacing humans as stand-up comedians [43, 49] and actors [35, 36]. A portion of the success of these robotic performances is thanks to their appearance. Recent investigations have explored the performance capability of non-humanoid entities. Robot Improv Puppet Theater (RIPT) introduces an audience-controlled robot arm that improvises short scenes using a variety of pre-programmed gestures [43]. Rond et al., explored the creative potential of a cylindrical mobile robot in improv situations as it moved through space to interact with performers [49]. In both cases, these studies successfully demonstrated the communicative potential of performance approaches to HRI through non-anthropomorphic robots that implement performance philosophies. While challenging, the research findings from both papers suggest meaningful interactions are possible with varying form factors in live performance scenarios.

Previous work in HRI has expanded the definition of a performer to include robots of various shapes, sizes, and proportions on stage [23, 38, 49]. Virtual personal assistants like Siri and Alexa can tell stories or beatbox on command. With these examples, the illusion of liveness falls squarely on the shoulders of the human interacting with the technology. Currently, most drones deployed for large-scale entertainment (concerts, light shows, etc.) rely on aerial programming to execute complex flying choreography and the performer-collaborator creates the sense of *liveness* or autonomy within the drone.

Another aspect of HRI performance practice exploration is the strength of live performance spaces sites for experimentation and discovery. For example, theatrical play creation introduces students to various robots; it gets them thinking about the philosophical implications of technology, asking them to develop a play where the robots are the main actors [29, 57]. However, in all examples, the degrees of freedom (DOF) these robots express are limited to physical expression. While several models from performance practice use generative exploration to iterate physical and gestural language, at best, any "liveness" in a robot or drone in these scenarios is the by-product of a transference, where the human participant (actor or bystander) attributes characteristics and expectations based on their understanding, experiences, or emotional state on to the drone [58].

Our relationship with robots constantly evolves to reflect their function, action, and capability. This rapid evolution requires a malleable practice to evaluate interactions. Drones present an exciting development in robot evolution due to their ability to fly. However, while the drone presents freedom in its ability to fly and interact, there are still pre-set conditions to navigate when using a drone, regardless of the interaction scenario. Challenges may arise

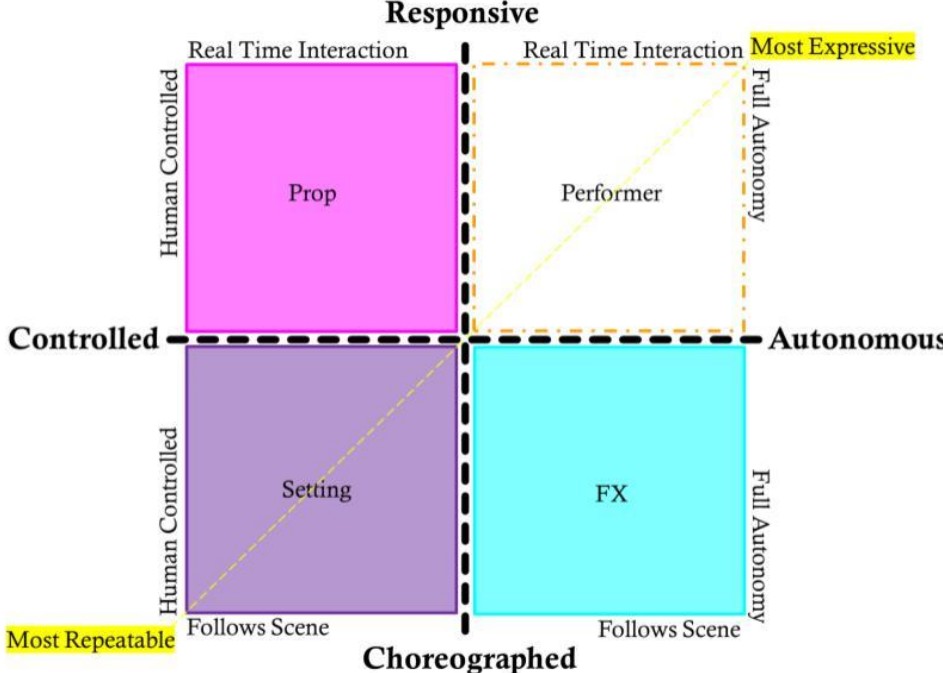

Figure 2: Four performance domain categories separated by their level of autonomy and responsiveness, with the top right quadrant "Performer" being the most interactive and expressive and bottom left quadrant "Setting" being the least interactive and expressive.

as HDI researchers negotiate and explore how humans perceive and respond to encounters with flying robots in different scenarios (emergency response vs recreational use) or applications [27]. One-way researchers have started to investigate the social potential of drones is through philosophies such as somatics [10], somaesthics [39], and techniques like Laban Movement Analysis (LMA) [41, 43]. More traditional performance spaces are also increasing in these kinds of drones [3, 4, 48]. Yet, the number of scholars citing entertainment applications in Herdel et al.'s survey [27] suggests a more profound interest in this novel space. We suggest that continued exploration of movement and other dramatic arts practices could greatly benefit HDI research.

## 2.2 Drones and Stage Craft

Where previously we reviewed existing examples of performance practices as tools for HDI and HRI research, here we further our explorations from a performance perspective. In the following section we present and define performance domain categories, familiar to theatre professionals, which are the technical elements of theatrical stagecraft [23, 24]. We then propose a new structure that arranges each domain based on the relative level of *autonomy* and *responsiveness* when engaging with human interactors, which we apply through an adapted version of Baytas et al.'s autonomous social drone four quadrant plot [12]. We use [12] given its significance to the field of human-centered autonomous drone interaction. However, we adjusted [12]'s "teleoperated to autonomous" *x-axis* to capture a broader spectrum of human intervention and drone autonomy by replacing "teleoperated" with "controlled". We also designed our *y-axis* to reflect the range of interactivity present in HDI research and the entertainment industry (figure 2).

We identified the qualities of the four main stagecraft categories and arrange them from *controlled* (requiring human intervention to operate) to *autonomous* (operation without human intervention) and *choreographed* (adherence to a specific sequence of movements through space) to *dynamic* (environmentally and situationally responsive). Figure 2 depicts the four major domains

we explore within live performance: *Setting, Prop, FX*, and *Performer*, and outlines their qualities and classifications. We categorize each domain field based on its respective level of responsiveness and autonomy. We chose these domains for their prominence within performance spaces and ubiquity among performance professionals. Furthermore, the domains we present act as containers for multiple types of interactions as explored in the following subsections.

To understand how our structure works and where drones can exist in performance scenarios, we must first define the various elements that make up our framework and create a live performance. Scenography is an umbrella term that is often broadly applied to three of four of our presented categories, encapsulating lighting, sound, costumes, and scenic design. It is mainly responsible for creating pictures and atmosphere on stage [23, 24]. In the following section, we divide the performance domain into four main categories: *setting, FX, props*, and *performer* and provide examples of how each category is implemented in traditional performance spaces. Finally, we suggest examples of contemporary HDI research that correspond to category to demonstrate how drone integration could bolster engagement during live performances. While not an exhaustive list, each category presented here reflects a significant element involved in the final product of a live performance, regardless of the production's scope, scale, and vision. The following subsections present a broad spectrum of performance interactions with examples relevant to contemporary performance. We suggest these categories as containers for current and speculative HDI research, as demonstrated through the inclusion of references from the HDI community in the following subsections.

### 2.2.1 Setting

For our purposes, we separate scenography into smaller components that help articulate application areas. Therefore, we define the *setting* as any elements that involve the design and construction of the actor's tangible environment to establish a physical location and time [23, 24]. To this end, our definition

includes the set itself (walls, doors, windows, etc.), set dressings, furniture pieces, and any other sizeable physical object visually presented in the performance space. A performer might interact with set pieces, but the set piece does not change location due to the interaction. Therefore, the setting is *controlled* (or passive in engagement) and *highly choreographed* (sticking to a specific topography).

Despite current limitations around drone size and payload capacity, Nozaki [45] has effectively demonstrated the potential of tandem drones equipped with a projector and a floating wall that could serve as performance spaces and moving set pieces. For example, projection-enabled mobile walls could allow location changes by implementing mobile static images or live video elements. Additionally, set pieces attached to drones could be reconfigured to create unique environments for characters to explore on stage.

### 2.2.2  Technical Effects (FX)

In live entertainment spaces technical elements such as sound and lighting are often colloquially referred to as tech. To avoid confusion across research domains we refer to these technical elements as *effects*, or simply FX for short. For our purposes, FX is any element that supports live performance by integrating sound, projection, lighting, or special effects. Borrowing from special effects we chose FX as it is analogous to the shorthand used to represent specific technical elements such as lighting (LX), sound (SX), and projection (PX). FX elements are *highly choreographed* (adhering to a specific series of movements). While more autonomous than static set elements because FX can run the length of performance from a singular command, FX elements still rely on a pre-determined program and are, therefore, *highly controlled*. FX elements can be integrated diegetically (within the narrative reality) or non-diegetically (outside of the narrative reality).

While Nozaki's [45] billboard drones use projection and could thus be considered an FX drone, their integration of the floating wall as projection surface is what places it more in the setting category. Comparatively, if we return to an earlier example from our introduction, Rhizomatiks dance collaborations with ELEVENPLAY have utilized drones to illuminate performers, often diegetically [3, 4]. Integrating the drones in this way provides opportunities for dancers to play with light, shadow, and shape and could potentially allow for more exploration and freedom of movement from the performers.

### 2.2.3  Stage Properties (Prop)

A *prop* is any object an actor engages with or physically moves across the stage, regardless of whether said object finds a new home or returns to its original position [55]. It is the interaction with the performer that gives the prop life. Props tell us something about the characters through their use and interactions; one character might have a cane, while another is rarely without a drink in their hand. Because of their inanimate nature and reliance on human interaction for any liveness or activity, props are *highly controlled*. However, props are also dynamic, as they can be *highly responsive* depending on their use. For instance, when one character throws their beverage in someone's face, or another draws a sword from their cane. A sword fight is highly choreographed to ensure safety. The sword itself is void of life until picked up by the performer, but any deviation from choreography could result in serious injury. Similarly, less dangerous props like a telephone spring to life when a technician makes it ring by pushing a button somewhere off stage; while less lethal, an error in the timing of a phone call can throw off the actors and production.

One could argue that most drones fall into this category as they tend to require a human pilot to initiate interaction. However, research such as *Drone Chi* [39] presents a mini drone disguised as a flower. As the flower drone enacts its flight path it also responds to the *proximity* (Section 4.4) of the participant's palms resulting in a reciprocal relationship to keep the drone's flight path as smooth as possible. The continued flight is a direct result of the active engagement of the participant maintaining active motion and appropriate proximity from the flying flower.

### 2.2.4  Performer

Perhaps the most nebulous category given the current state of HDI research is that of the performer. Performances are the result of rigorous repetition and reflections in the rehearsal process. The merit of an actor is their ability to execute a final performance that is live and responsive to the moment. If acting is reacting, as renowned acting coach Stella Adler suggests [6], a performer must be *responsive* in their interactions on stage, whether with their scene partners or the environment. The best actors can repeat their blocking and recite their text with precision while remaining responsive to their fellow actors and the life of the scene. Concurrently, their partners do the same, resulting in *autonomous*, *dynamic* exchanges onstage.

While *Dancing with Drones* [21] and *Aeroquake* [31] present unique methods of integrating drones as visible onstage "performers" and characters, the challenge with categorizing drones as performers remains their lack of autonomy. In both cases any semblance of liveness in the drones is the responsibility of the human performer in the interaction. The performance capabilities of drones are rooted in a drone's ability to participate in the act of creative expression and extend a drone's ability to communicate with human bystanders while completing a task-based action. A common factor when deploying drones in performance and social scenarios is the necessity to communicate with a human partner.

What both *Dancing* [21] and *Aeroquake* [31] have in common is their use of movement as a communication language. Movement remains one of the primary modes of communication in HDI research, as seen in the examples provided throughout Section 2.1. Movement communication practices in HDI can be linked back to the performing arts as Rudolph Laban's Laban Movement Analysis (LMA) [1], a foundational approach to movement in performance work, permeates the HDI domain. Throughout Section 2 we highlighted examples of performance techniques and approaches within HDI demonstrating the value and prevalence of transdisciplinary approaches. Using this same approach, we offered a classification system for drone deployment based on live performance categories. In the next section we will further explore LMA as a tool for exploration, communication, and compositional movement.

## 3  LABAN MOVEMENT ANALYSIS

Laban/Bartinieff Movement Analysis (LMA or Laban Technique) [1] has been frequently cited in HRI to define movement qualities for communication [9, 42, 43]. LMA has become a reliable tool in HDI for expressivity through movement in anthropomorphic and non-anthropomorphic agents. The introduction of LMA to HDI resulted from a collaboration between researchers and performance professionals that use flight path as a communication tool [9, 41]. We explore LMA through the Eight Efforts (Table 1), a series of actions with specific qualities that can be used to communicate intention or internal emotional states [1]. We couple this intentional movement practice with the introduction of compositional movement system, Viewpoints [8, 14]. The pairing allows us to integrate generative and iterative movement practices into our exploration.

### 3.1  The Eight Efforts

LMA consists of four categories: body, shape, space, and effort. Body is concerned with how the physical self works on an internal

| ACTION | WEIGHT | SPACE | TIME | FLOW |
|--------|--------|-------|------|------|
| Punch | Heavy | Direct | Quick | Bound |
| Dab | Light | Direct | Quick | Free |
| Press | Heavy | Direct | Sustained | Bound |
| Glide | Light | Direct | Sustained | Free |
| Slash | Heavy | Indirect | Quick | Bound |
| Flick | Light | Indirect | Quick | Free |
| Wring | Heavy | Indirect | Sustained | Bound |
| Float | Light | Indirect | Sustained | Free |

Table 1. Laban's Eight Efforts

level, including the connection of limbs and the origin of movement within the self. Shape investigates how the body changes in space while producing movement. Space is the external environment. Finally, effort is the intentionality of the movement through physical exertion. These four categories form the basis of movement exploration. Beginning internally with sensations within the body and becoming increasingly external with each layer as the performer explores how to express internalized feelings by creating shapes with the body, how the expression of shape moves the body through space, and how effort impacts how the performer sees the space they move through.

Laban's movement practice is incremental and explores the polar qualities of movement defined as space, weight, time, and flow (Figure 3). For example, a move through space can be either direct or indirect, whereas weight can be light or heavy. In addition, the time it takes to complete a movement can be either sustained or quick, and the flow of an action can be bound or free. These polar qualities make up the basis of Laban's Eight Efforts (Table 1). The Eight Efforts create spectrums of movement that performers can utilize to help differentiate the types of movement in various combinations. Combining the Eight Efforts results in qualities of movement that have been effectively applied to gauge intention in non-verbal HDI [51].

Actors utilize Laban's Eight Efforts for character creation and communication in traditional performance spaces through physical language. Laban's continued and layered movement exploration is incremental, allowing time for investigation and reflection. Laban's Eight Efforts also supplies a shorthand of language that captures the combination of these polar movement qualities. Table 1 depicts the Eight Efforts, a chart that assigns a single verb meant to capture and define the qualities of the combined movement factors exhibited. These verb keywords (or Actions) make it easier to communicate multiple sensations with a singular word. For example, a director might ask a character to *Float* across the stage, meaning they want the actor to move lightly in an indirect way, sustaining each free-flowing step or gesture. Alternatively, a performer may be playing a character that *Punches*, or moves in a very heavy, direct, quick, and bound way.

When applying Laban practices to drones, it is vital to remember that LMA is still rooted in understanding human movement. Qualities like time or space seem easily transferable. In contrast, the expressive light and heavy (weight) or bound and free (flow) qualities may be more challenging to discern in flying non-anthropomorphic forms. Therefore, a new language is required to

---

[1] In Brook's famous "shoe," example, actors from different cultural and linguistic background created work by interacting with a shoe borrowed from an audience member. These extensive explorations were conducted during a two-year research project

explore and discuss drones' unique and specific movement qualities.

### 3.2 Compositional Movement

There are many tools at an actor's disposal to communicate intention that ranges from the profoundly psychological [24, 52] to the physical [1, 8, 14, 32]. Viewpoints [8, 14] is adapted from movement practice by acting coaches, Bogart, and Landau. Viewpoints provides a system for developing nonverbal communication through iterative exploration and creation, which they call composition [8]. Viewpoints are concerned more with observation and evocative movement than emotion and is therefore helpful when dealing with less-emotive agents (e.g., robots). The shift to externally motivated interactions rather than internally motivated ones emphasizes exploration through improvisation and ensemble (team) building [13]. The Viewpoints approach to movement creation clarifies how physical investigations can lead to intentional and somatically motivated movements. Part of the strength of both Viewpoints and LMA is that both are internationally recognized schools of movement, meaning, much like HCI/HDI, they can be applied in various cultural contexts. We propose that understanding physical communication with live performers can be transferred to drones and support a more thorough analysis of contextual movement when combined with Laban's Eight Efforts [1] (Table 1), which we explore in the next section.

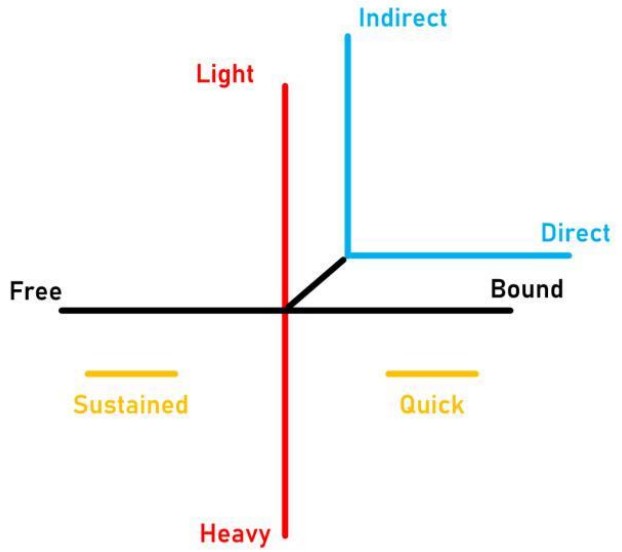

Figure 2: Qualities of Laban Movement Analysis.

### 4 PERFORMANCE WORKSHOP AND EXPLORATION

Our research explores the terrain between HDI and performance practice to create new work based on the relationship between humans and drones in a design environment. To conduct this work, we ran a Performance Workshop (PW) testing our framework as a method of performance creation and non-verbal communication. In performance, the PW is the primary research method involving intensive study over a single, extended period. Following the work of Peter Brook[1] [15] and Keith Johnstone[2] [30] on the interaction

---

funded by the National Theatre and were recorded in The Empty Space.

[2] Extending work from Peter Brook, Keith Johnstone shares the results of multiple experiments using real and imagined objects as

between actors and objects as codified in the theoretical work by scholars such as Shomitt Mitter[3] [44], the PW iteratively tests performance objects and refines them until they are production ready. Subsequent "testing" involves the presentation of this work to an audience, but the primary research model is focused on a detailed study of the individual performer interacting with objects in a designed environment.

Performance workshops involve three distinct phases: 1) Tuning; 2) Discovery and Iteration; and 3) Production. In the first phase the team works to harmonize their approach to the work. This work is like the members of an orchestra tuning their instruments before a performance but requires several hours during which performers conduct a series of exercises designed to serve the objectives of the performance workshop (in this case a study of human-drone relationships) and the means of working (in this case Laban Movement Analysis). In the second phase, the group rehearses sequences of work to note questions and discoveries, pausing to develop those that stand out to the group. Items that prove fruitful are given extra time and run repeatedly until the group feels that the area has been exhausted. In the third and final phase, the group returns to the items developed during phase two and using the performance workshop objectives and approach runs the segments to get them "production ready." Production ready means that the work can now be demonstrated within a professional environment for potential use in a live show. At that stage, production staff will evaluate the safety implications and technical requirements, directors, actors, and writers will be consulted to prepare for their integration in live work, and the venue for the work will provide requirements for the use of the work.

Performance workshops are necessarily small. Often using a single performer, this work requires intensive focus with detailed cataloguing of problems, questions, and potential. In this study, we used three performers to triangulate our work, while preserving the need for focused exploration. Performance studies move to larger numbers when they expose their work to preview audiences. The feedback from these early audiences is crucial, but it arrives just before professional production, so the performance workshop is designed to refine the performance objects so that only minor alterations are required before public presentation.

## 4.1 Methodology

Our project utilized qualitative methods familiar in drama, performance, and human-drone interaction research. A qualitative approach allows for ideation and performance generation alongside reflective analysis. Our explorations are based on a tradition of embodied practice **Error! Reference source not found.** familiar to performing artists. Like somaesthetics [10, 28, 39], embodied practice is focused on the connection between the body and the self.

To frame our workshop, we recorded the session following the PW approach, including participant-led discussions between each performance exploration. After the workshop, video and audio files were transcribed. All discourse was assessed for content and theme and compared to the corresponding physical explorations using open coding, grounded theory [13, 46, 54] and interaction analysis [7]. Combining qualitative methods allows a thorough examination of an extensive data set and supports the inherently interdisciplinary nature of HDI, especially in the arts and entertainment. Reviewing our discussions revealed specific dynamics between the performance professionals defined by the specializations and relationships they brought into the session.

We recruited two professional ballroom dancers (P1 & P2) with 40 years of combined experience. The participants were selected from a network of local performance artists, due to their familiarity with each other from years of competitive dancing as a couple. Their familiarity eliminated any need for introductions, ice breakers or performance translations outside of those with the workshop material and the facilitator. Neither participant had any previous experience or direct interactions with drones before the workshop.

The primary author acted as the workshop facilitator due to their extensive background as a performing arts professional which includes over 20 years as a director, actor, and theatre technician and extensive work with both Viewpoints and LMA. The facilitator led exercises, and discussions, without participating in the activities. The workshop lasted approximately three hours with breaks between each exercise for discussion.

Given the wealth of experience brought to the workshop by our artist collaborators, we relied on embodied practice as it is embedded in the studio learning process and knowledge transference, where performers develop their art through observation, application, and repetition of the techniques they encounter. Framing our exploration as a workshop was then not only a familiar way of working for our dancers but took advantage of physical and gestural language. Furthermore, we supported our exploration with the Critical Response Process (CRP) [40]. CRP is a practice that stems from professional dance and has been adopted by other performance practitioners and artists to provide and receive feedback on artistic works in progress. CRP is a valuable tool in artistic practice as its primary focus is the artist's intention. CRP provides space for the audience, or in our case, the session facilitator, to engage directly with the artists to understand and give feedback that helps the artists clarify their intentions. Through our use of CRP, we were able to better communicate and iterate with our dance collaborators. We felt that it was essential to explore the potential applications of drones in this new space, while addressing resources and information from drama and HDI.

In the following section, we review our workshop in two sections, one dedicated to the introduction of Laban Movement Analysis and the second to the introduction of our drone stand in. We separated the workshop into two parts to help reduce the cognitive load on the participants while introducing them to new concepts, techniques, and technology.

## 4.2 Workshop Part One: Introduction to Laban

Participants were welcomed to the lab and introduced to the research team before being led through short warmups designed to familiarize them with the specific kinds of movement and physical thinking they would be asked to do. The participants were then guided through various Laban movement exercises in which they explored Laban's Eight Efforts [1] while moving through space. As the concept of LMA was new to both participants, the facilitator introduced prompts to help the participants work through the sensations internally and externally. For instance, when exploring the *heavy* quality of weight, we asked questions such as: "What does it feel like to move through the space in a heavy way?"," Where does that heaviness live in the body?", "Can the sensation of heaviness be transferred to another part of the body?", "How does moving it change the way you move through space?", "What happens if you externalize the sensation of heaviness as if the room is filled with honey?" etc.

---

the driving force of performance moments in the context of improvised scenes performed live.

[3] Mitter sets out his study for formal performance practice by celebrating the work of Brook, Johstone and others (such as

Grotowski), but calling for formal theoretical study that allows for the use of their work in academic and professional practices p.5

The participants were instructed to adopt a neutral walk and move through the space, an approximately 6x6 M interview lab with the furniture and equipment pushed to the edges to allow for larger movements. The goal of the neutral walk is to actively identify and release any tension held within the body while moving without intention. The introspection that comes from a neutral walk serves as a base for movement exploration generated by specific prompts. Starting with *direct*, the participants were instructed to introduce Laban's movement qualities into their walk and reflect on how the quality impacted the way they moved through, viewed, and engaged the space. After several minutes of exploration, the participants were instructed to return to a neutral walk before introducing an opposing quality. In the case of our earlier example, the polar quality to *heavy* is *light* movements, and the prompts were repeated. Each exploration was followed by a brief discussion regarding the somatic sensations [28], challenges and discoveries of each movement quality. Once given a chance to work through all the individual movement qualities, the solo exercise was repeated with the participants combining multiple qualities to explore the Eight Efforts (Table 1, Figure 3).

### 4.3 Workshop Part Two: Introduction to Drones

In the second half of the studio session, a non-functional foam body drone was introduced and given to P2. The exercises from the morning were repeated with P2 instructed to imbue the drone they were 'puppetting' with the given qualities. We chose manual intervention over flying the drone for the second half of the workshop for several reasons:

1) to make the interaction safer by not having a drone flying in the confined lab space.
2) Puppeting eliminated the steep learning curve of flying drones remotely.
3) Having P2 puppet the drone liberated the participants from the actual drone functionality, allowing them the freedom to explore how they would like to see drones move in a live performance with a human.

As in the first half, P1 focused on embodying Laban's efforts. Once again, both performers were led through the same series of exercises, starting with exploring the individual movement qualities. Working through the qualities again gave the performers time to familiarize themselves with the newly introduced element, the drone. Once all individual qualities were explored, the qualities were combined, and the performers were again led through an exploration of Laban's Eight Efforts. Now more familiar with movements, terms, and efforts, the participants were again guided through the Laban exercises. This time, they were encouraged to explore how movement qualities impacted their interactions as a pairing and the cycle restarted. P2 was again given the drone and instructed to investigate how the drone might manifest the movement qualities and actions.

### 4.4 Findings

Both performers worked well within Laban Movement Analysis with each finding an effective to way internalize the exercises. For example, when reflecting on moving in a *direct* way, P1 said they felt like they were "moving with purpose," while P2 felt like they "had lost something and were trying to find it." While the quality was the same for both participants, the intentions were unique to the individual resulting in two distinct physical manifestations. This success continued when the drone joined the performance.

During the process P1 continually wanted to get close to the drone as if to inspect it. In doing so, they became aware of the miniature camera lens in the nose of the drone. In our post-session reflection, P1 reported feeling "watched" by the drone, feeling like they were "performing movements for the drone to observe". Despite the voyeuristic quality (which could be taken positively or

negatively depending on the conditions of the drone's deployment), P1 remarked that "the closer the drone was, the more familiar it felt." This effect increased after extended periods of exposure. The recordings show that P1 actively explores and even plays with the distance between themselves and the drone. This continued play with distance and personal space motivated our contribution of *proximity* as an extension of Laban's Movement Analysis framework (Figure 4, Purple). The polar qualities we attribute to *proximity* also come directly from P1's explorations, with *intimate* an expression of their comfort and familiarity and *aloof* being a term they used to describe their desire to "play hard to get" with the drone. In this approach P1 clearly demonstrated an invitation to the drone to engage. P1 played *with* the drone, but when the workshop moved to P2 the performer tended to play t*hrough* the drone.

P2, the drone "puppeteer," noted that while manipulating the drone, their focus remained on P1, likely resulting from their pre-existing work as dance partners. In terms of movement, P2 maintained a relatively stiff or formal posture that was distinct from P1s more playful approach. While P2 did not note any significant physical shifts in their approach to drone movement, the workshop recording reveals that: 1) As a result of their physical form, most drone movements were expressed and articulated through the wrist and 2) P2 tended to move the drone on a vertical axis, either crouching low to the ground or extending their hand high up in the air to observe their performance partner. When asked about these vertical movement choices P2 responded that they were "attempting to express a sense of flying." These observations formed the second of our drone-specific movement qualities, *height* which fits within the existing LMA schema (Figure 4, Green). That the differences between performance styles still fit within the Laban

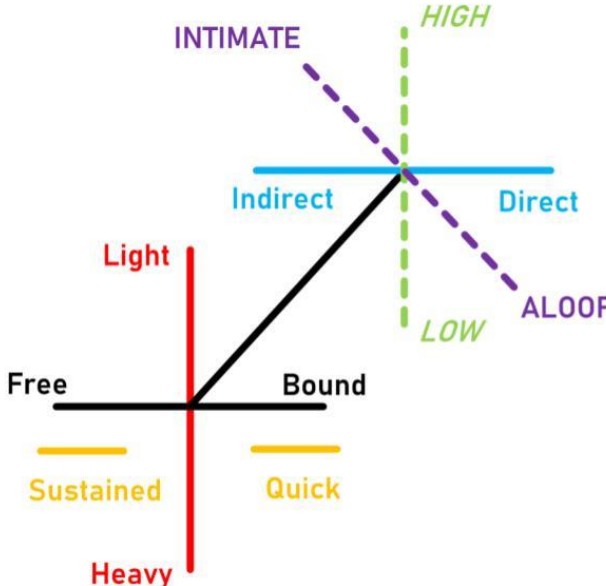

Figure 3: Extended drone specific Laban framework.

Movement Analysis reinforced our confidence in LMA as a formal structure that integrates performance, drone interaction and the overall communicative environment providing crucial information for production.

## 5 DISCUSSION

There are typically two ways to approach data collection in the performing arts that reflect the typical qualitative/quantitative dichotomy found in interaction research. Quantitative performance studies reflect on the experience of an audience, allowing for the

collection of a larger data set focused on a final product (performance). Alternatively, qualitative inquiries focus on the experience of the performers. While a comparatively smaller test bed, working directly with the performers provides a deeper and richer analysis of process and final product. Here we discuss the implications of our work including the limitations of our exploration and our hope for potential future works.

The intention of LMA and many other movement practices is the expression that stems from internal exploration and reflection. One way to integrate drone-specific movement into the existing Laban framework is for height and proximity to replace weight and flow (Figure 3), as we found the latter two challenging to discern in a flying object. Another option is to extend the qualities of movement to include drone-specific movements. In Figure 4, we propose a comprehensive framework incorporating drone-specific movements as part of the spatial spectrum of LMA as they deal specifically with the drone's physical self. Expanding the framework in this way allows researchers and performers to capture the unique flight capabilities of drones while still allowing for explorations of weight and flow.

For our qualities to express the full range of potential drone movement, we propose adding *height* and *proximity* (Figure 4), new categories of unique drone movement qualities to the existing LMA framework. *Height* and *proximity* are factors that impact the spatial relationships of drones to the surrounding world and can be easily understood by human interactors. Furthermore, height and proximity address a flight, a movement capability unique to drones in HRI and the human social concern of personal space when encountering an unknown flying robot in public settings. *Height* is directly connected to a drone's ability to fly and is intended to explore the perceived difference between being high in the air or low to the ground. Rapid changes in height can communicate distress or despair, as seen in *Dancing with Drone's* operatic exploration [21]. Conversely, a gradual decrease in verticality could indicate a low battery or fatigue [17]. *Proximity* refers to the spatial relationship between the drone and human interactor, where we define *aloof* as a distant interaction, such as bystander observation; and Intimate as any interaction within one meter of the person's personal space [25]. We observed that as P1 and P2 worked through all various combinations of Actions and drone movement qualities (*aloof* became *intimate* and *high* became *low*). While P2 was exploring the Actions through the drone, they were also responding to P1's physicality. The longer the participants were allowed to explore the physical qualities presented the more comfortable they became. The more comfortable they became, the more they began to converse with each other through movement.

Thus far, we have attempted to make a case for the power of performance practices, specifically movement in HDI. Our performance workshop explored a movement practice familiar to performance and HDI domains. In addition, we attempted to contribute new terms to existing language that could help support interaction research by integrating performance practice. However, it is essential to remember that this collaboration can go both ways and that several examples of HDI research and technology can be implemented in performance spaces.

## 5.1 Limitations

Our preliminary workshop leveraged the over five decades of experience brought by our two participants and our facilitator over a three-hour afternoon. However, the strict focus of the participant's practice as ballroom dancers came with limitations as both participants displayed a tendency to fall back into familiar postures and patterns from their dance training. While dance is a highly physical language designed to communicate nonverbally, many styles of dance are based on a core foundation of movements. In broader performance traditions, movement practice is its own

discipline populated by experts with over 10, 000 hours of dedicated studio practice in specific systems which may require an understanding of the fundamentals of that specific practice to encourage exploratory movement.

Additionally, the lab studio did not lend itself to full range of motion exploration. The collaborators had to negotiate the hard tiled floor while navigating furniture pieces in the room. Collaborating with LMA specialists in an environment more conducive to performance exploration in a more long-form exploration will help us refine our preliminary findings.

Finally, while the workshop format lends itself well to this kind of exploratory creation, execution relies solely on participant expression and LMA is designed to be interpreted by the individual as described in Section 4.4. Performance workshops are also designed to be intensive and iterative. Therefore, the workshop format, like LMA, is repeatable but not replicable. Further explorations with different performers, or even the same performers on a different day will yield different interactions and insights. We hope that through collaborative development and composition, we will be able to further explore the implications of *proximity* and *height* to create a more concrete movement analysis method that addresses the needs of both the live performance and HDI communities.

## 5.2 Future Work

Laban Movement Analysis is far from the only movement-specific practice in dramatic performance. Anne Bogart and Tina Landau's Viewpoints [8, 14] provide an extensive system for understanding physical communication and composition with live performers that can be transferred to drones and support a more thorough analysis of contextual movement when combined with Laban's Eight Efforts [1]. Viewpoints are concerned more with observation and evocative movement than emotion and are therefore helpful when dealing with non-emotive agents (robots). Where Laban Movement Analysis presents four efforts as categories of physical expression, Bogart and Bartinieff use nine physical viewpoints separated into two main categories, space, and time [14]. The nine viewpoints result in a vocabulary capable of producing a somatic lexicon from which data sets can be derived to help define, understand, and create movement in participants. When used in conjunction with Laban Movement Analysis, researchers can develop a broader language to discuss the qualities of movement. For example, if we consider the physical form factor of drones as their "body," we can better categorize and analyze the movements of both human and drone participants in HDI settings.

The framework we presented is only one piece of the LMA equation; the combination of movement qualities into what Laban calls efforts or actions forms the other (Table 1). To further gauge the application of our dynamic performance drones, subsequent high-level performer studies with professionals across disciplines, i.e., dance, drama, and theatre design are necessary. We believe that bringing in a larger group of professional collaborators to explore the potential applications within a dedicated performance space for performance-specific domains is essential for further developing movement vocabularies for dynamic drones. Doing so will allow us to redefine the actions of Laban's Eight Efforts. By implementing our height and proximity efforts, the result is an expansion and redefinition of Laban's Eight Actions that we believe will benefit the future creation of a drone-specific movement lexicon for use in both performance and broader research scenarios. Furthermore, our workshop was purely exploratory, applying our findings in a performance scenario is still required to determine the effectiveness of performance drones onstage. Deploying drones in a live performance scenario will test their performance in show conditions while allowing us to conduct quantitative audience

impact studies to gauge the reception of drones and clarity of communication within a larger performance context.

Disney theme parks have applied for patents that utilize drones in large aerial shows and as mechanisms to control giant inflatable mascots like marionettes [19]. Puppetry is unique in performance practice as it brings inanimate objects to life and creates emotional connections through expression via precise movements. There are many correlations between the rich tradition of puppetry work and performance that can be applied to drone operations. While we did not touch on it in this paper, there is vast potential in integrating practices and philosophies from puppet-specific performance domains such as marionette [18, 33] work or Japanese bunraku [20]. Further exploring the possible connections between puppetry and drones will undoubtedly result in more immersive, gesture-based drone controls that could allow for precise actuation by a human pilot and address questions of liveness and dynamic interactions on stage. Such practice can be found in [39], a drone-based practice for tai-chi. Despite it not being part of the puppetry domain, it uses refined physical practice to follow the same principles of precise actuation by a human pilot and can be adapted to performance arts. In the next section, we expand on some of the examples provided by presenting the different domains within live theatre and explore how drones might be used to evoke a sense of magical realism onstage.

## 6 CONCLUSION

Drones are becoming more ubiquitous across a variety of domains. As they continue to expand into social spaces and engage more directly with humans, researchers must continue to find ways to communicate desire and intention to reflect the drone's varied applications.

This paper explores the overlap between performance applications, practice, and HDI research. In doing so, we introduced a new design space for drones' deployment in live performance settings demonstrating that with a little lateral thinking, current drone applications also fit within the four performance categories we identify: *props, setting, FX,* and *performer.* Furthermore, we demonstrated the potential for creativity in HDI by exploring performance methodologies in our pilot performance workshop. Leveraging the explorative potential of the workshop format we were able to devise two new drone-specific movement efforts, *height* and *proximity,* and qualities, high/low and intimate/aloof, respectively. We contribute these efforts and qualities in a new drone specific movement framework that expands on established movement practices resulting in a collaborative model that can be used for physical communication.

While our primary goal is the exploration of models for application in live performance, we feel that with the continued experimentation with larger, more diverse performers could yield positively impact the research and development of future HDI social robotic systems. Better communication and expression from drones can affect everything from delivery drones to remote first responders. In addition to collaborating with performers, we look forward to further developing our framework in more traditional HCI research environments with HDI and social robotics experts. We hope that more researchers will realize the potential of performance practices inside of HDI and adopt transdisciplinary approaches to HDI research. As demonstrated in this paper and supported by previous investigations, cross-domain collaborations can help develop expressiveness in robots and drones and provide unique interactive perspectives capable of generating rich, reflective data. From gestural interfaces and nonverbal communication to exploring how humans might perceive, use, and coexist with flying robots in the future, drama and performance domains contain a wealth of knowledge that is of great use to HDI research and is only just beginning to be explored.

## ACKNOWLEDGEMENTS

This research is the result of a collaboration between The University of Calgary and Ben Gurion University of the Negev and was funded through MITACS Globalink Research Award.

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
