# OpenReview forum: "Waiting in the Wings: Drones in Live Performance"
_graphicsinterface.org/Graphics_Interface/2023/Conference_SD — GI 2023 - second deadline_

### Official Review · Reviewer_4e2R · 2023-04-20
**Human Drone Interaction Study**

**Rating:** 4
**Confidence:** 4

**Review:**

This article explores how to fully integrate drones into live performances and practical lessons learnt. I find the idea very interesting and exploratory. However, the work is not ready at this stage for publication.
The paper attempts to contribute to the development of a theoretical framework of new movement techniques and vocabulary for experimentation with dynamic drone actors. It is rather unclear how the framework is made and what is innovative about the framework compared to the existing literature (e.g., LMA).

The related work section is limited. The structure of this section needs improvement. Also, RW is scattered in the paper, some are in Section 2, and some are in section 3.2 – I suggest combining them.
Section 3 – The performance domain divisions are a great idea; however, the names are a bit confusing; , the names chosen for these divisions are not representative of the explanation provided later. The four stages are not well explained when introduced; my understanding is that these stages are defined by the authors. Is there any literature supporting these divisions? If yes, it would be great if it can be added and explained.
Some subsection (stages) discuss their relation to drones, some do not. For example, how does the Setting relate to the drone interaction?
It is unclear how these divisions (Fig.1) will be used for the proposed research.
Similar comment for Section 4 – I am not sure how the descriptions provided in this section are used in the study.
The study explanation lacks the detailed information necessary to assess the validity of the study. For example, information about the recruitment process, participants’ demographics, …
The study is conducted with only 2 participants, additional rationale behind the reason 2 is sufficient must be added to the paper.
Although the findings of the study and the discussion are interesting, it does not directly relate to how these findings can be used in the field of HCI.

Overall, this is an interesting topic that is explored by the authors, some parts of the paper need further explanation and the results need to be discussed in relation to the HCI community.

Minor suggestions:
The abstract can be expanded to include more details about the results.
Section 2 – the title should be edited.
Section 3 – In the first paragraph, FX should be defined.
Section 3.1. the order of subsections (stages) does not align with Fig 2.
Section 5.1 - cite Merce Cunningham is missing.

---

### Official Review · Reviewer_44jx · 2023-04-23
**Interesting submission, on the fence about acceptance**

**Rating:** 5
**Confidence:** 2

**Review:**

In this paper, the authors present a framework to look at HDI research from an artistic viewpoint.
I found the paper a very interesting read and a timely one as well.
Before I dive further into my analysis of the paper, I would first like to note that I am not an expert in artistic viewpoints or theories that the authors bring forward in this submission. I will therefore not be able to weigh on these theories and I hope that another reviewer can.

I am quite confused by the structure of the paper. For instance, section 2 appears to be the related work and goes into a subsection 2.1 and then a subsubsection 2.1.1 without ever having a 2.2 or a 2.1.2. The paper then jumps to a section 3 without clearly establishing a summary from the related work which I also found surprising.

The subsection 2 is also quite peculiar for me and I am unclear as to how much of it is interesting or important for the paper's argumentation. Indeed, as it is, it seems that 2.1 only lists examples of performances (which could easily be in the introduction). It is also unclear how much of section 2 overall simply re-represents the findings of [1]. I am not saying here that the authors are paraphrasing this systematic review, but if they are it should be made clear. Similarly, much of section 3 reads as a related work section and, while it is an interesting read, it is not always clear what I, as a reader, am supposed to take away from this text.

Section 4 and 5 are particularly informative and interesting to read and the workshop methodology employed by the authors is well motivated and makes sense considering their goals. Being unfamiliar with running workshops with such few participants, I am wondering how robust these findings are. I am sure that they are a good starting point but I wonder if they would translate to other performers in the same way.

Similarly, the authors state that they "propose live performance applications as a new design space for HDI research." but the paper only focuses on a specific set of art performances and stating that their design space would apply to HDI research as a whole might be pushing it. Reading the paper, I do not think that they have made a strong argument in this direction and I would therefore argue that the paper might be overstating its contributions.


Overall, I really found this paper to be an interesting and enlightening read. It presented me with a perspective and design space I was not aware of and I would argue that such contributions are likely to spark interesting and lively discussions at a conference. However, the paper, as it is, is also not really delivering what it set out to deliver I would tend to argue. So I am really on the fence. In a sense I want the authors to get a chance to spark discussions at the conference but I am unsure if the paper is good enough to achieve the required quality of the conference. My current score is therefore geared toward a neutral one, but I could easily be persuaded by someone with more expertise in these kinds of contributions than me. If anything, structural changes and rewrites should be done to make the paper clearer, it might be me, but I was really confused by some of the structure and I think this can be easily addressed.


Minor: the title of section 2 is probably in need of editing.

---

### Official Review · Reviewer_VpFd · 2023-04-24
**Exploring the Intersection of Performance Art and HDI/HRI**

**Rating:** 7
**Confidence:** 2

**Review:**

This work relates performance art to HDI research through a thorough literature review, detailed analysis of performance practices, and a qualitative study with two professional dancers. The findings from the study help to reconsider Laban's movement practice and evolve it in a drone-specific way, providing valuable contributions to the community.


The design procedure is introduced in convincing detail, and I found it entertaining to read the motivations, findings, and discussions, partly due to the lively style of presentation in this manuscript.

I believe this work provides illuminating insights for both performance art and HDI/HRI practitioners. However, the findings might be limited by the scale of participants, as they are based on the experiences and ideations generated by the recruited group of two dancers.

Additionally, the study setup of P2 holding the drone 'manually' limits the exploration of a multi-drone scenario, which is quite common in the field.

As a summary, I think GI'23 should accept this work with a rating of 7/10.


Minor issues:
	• #1: Based on the formatting guidelines, please sort all bibliographic entries alphabetically by the last name of the first author, rather than by appearance order.
	•  #2: There is an unresolved citation in Section 5.1.